# Towards Versatile Embodied Navigation

**Hanqing Wang**[1, 2]  **Wei Liang**[1,4*]  **Luc Van Gool**[2]  **Wenguan Wang**[3*]

[1]Beijing Institute of Technology  [2]Computer Vision Lab, ETH Zurich
[3]ReLER, AAII, University of Technology Sydney
[4]Yangtze Delta Region Academy of Beijing Institute of Technology, Jiaxing

Project page: https://github.com/hanqingwangai/VXN

## Abstract

With the emergence of varied visual navigation tasks (*e.g.*, image-/object-/audio-goal and vision-language navigation) that specify the target in different ways, the community has made appealing advances in training specialized agents capable of handling individual navigation tasks well. Given plenty of embodied navigation tasks and task-specific solutions, we address a more fundamental question: *can we learn a single powerful agent that masters not one but multiple navigation tasks concurrently*? First, we propose VXN, a large-scale 3D dataset that instantiates four classic navigation tasks in standardized, continuous, and audiovisual-rich environments. Second, we propose VIENNA, a versatile embodied navigation agent that simultaneously learns to perform the four navigation tasks with one model. Building upon a full-attentive architecture, VIENNA formulates various navigation tasks as a unified, *parse-and-query* procedure: the target description, augmented with four task embeddings, is comprehensively interpreted into a set of diversified goal vectors, which are refined as the navigation progresses, and used as queries to retrieve supportive context from episodic history for decision making. This enables the reuse of knowledge across navigation tasks with varying input domains/modalities. We empirically demonstrate that, compared with learning each visual navigation task individually, our multitask agent achieves comparable or even better performance with reduced complexity.

## 1  Introduction

As a fundamental research topic, visual navigation has attained extensive attention across many disciplines, including robotics [1], computer vision [2, 3], and natural language processing [4]. Consider a typical navigation scenario (Fig. 1), in which a human intends to direct a robot agent to navigate to a target – a buzzing washer. The target can be specified by a photo of the washer (*i.e.*, 🔍 ***image-goal nav.*** [5]), or the buzzing sound (*i.e.*, 🔊 ***audio-goal nav.*** [6]), or the corresponding semantic tag – *washing machine* (*i.e.*, 🏷️ ***object-goal nav.*** [7]), or linguistic instructions – "*go to the end of this corridor, turn left and enter the laundry-room*" (*i.e.*, 📄 ***vision-language nav.*** [8]). Naturally, the agent is expected to be smart enough to execute all these kinds of navigation tasks involving varying modalities/domains (*i.e.*, image, audio, semantic tag, text) with different optimal policies. Contrary to our expectation, almost all existing navigation agents are specifically designed/trained for one specific task – a "versatile" agent capable of mastering multiple navigation tasks remains far beyond reach.

Besides its great value in practice, investigating embodied navigation in multitask scenarios can help better understand human intelligence. First, we humans can learn multiple tasks in a parallel ad hoc

---

*Corresponding authors.

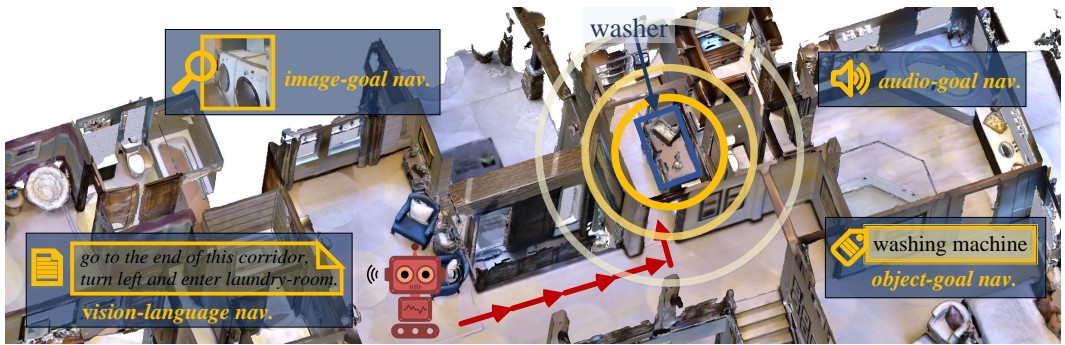

Figure 1: Rather than existing efforts training specialized agents on individual navigation tasks, we build a single powerful agent that can undertake multiple tasks, *i.e.*, 🔍*image-goal nav.*, 🔊*audio-goal nav.*, 🏷*object-goal nav.*, and 📄*vision-language nav.*, in visually and acoustically realistic environments.

manner, and benefit from commonalities across related tasks [9]. Second, we accomplish tasks by processing and combining signals from different modalities. Evidences from cognitive psychology indicate that our senses are functioning together and multisensory integration is a central tenant of human intelligence [10, 11]. Though the idea of multitask learning [12] was widely explored in computer vision field [13], prior attempts are often made in unsupervised and supervised learning settings; in the context of multitask reinforcement learning (MTRL) [14], not much is done for visually-rich, embodied navigation scenarios. One possible reason is the lack of a suitable dataset, compounded by considerable costs involved in data collection, as multiple navigation tasks should be supported.

In response, a large-scale 3D dataset, VXN, is established to investigate *multitask multimodal embodied navigation* in *audiovisual complex* indoor environments. VXN allows simulated robot agents to concurrently learn four tasks, *i.e.*, *image-goal nav.*, *audio-goal nav.*, *object-goal nav.*, and *vision-language nav.*, in continuous, acoustically-realistic and perceptually-rich world[2]. Based on a high-throughput simulator [15], VXN instantiates different navigation tasks in unified environments following the same physical rules. It equips the agents with multimodal sensors to gather information from 360° RGBD and audio observations. Taken all together, VXN provides a realistic testbed for multitask navigation.

With VXN, we further develop VIENNA, a versatile embodied navigation agent that jointly learns to solve the four navigation tasks using *one single model* without switching among different models. Based on Transformer encoder-decoder architecture [16], VIENNA encodes the full episode history of multisensory inputs (*i.e.*, RGB, depth, and audio) and navigation actions, and absorbs common knowledge across different navigation tasks with a shared decoder. Target signals (*i.e.*, goal picture, target class, aural cues, linguistic instruction) are parsed into *queries*, and the supportive context retrieved from the encoded history is fed to corresponding *policy* for task-specific decision making. With such a *fully-attentive* model design, VIENNA is able to comprehend multimodal observations, conduct long-term reasoning, and, more essentially, exploit cross-task knowledge.

By contrasting our VIENNA to several single-task counterparts on VXN, we empirically demonstrate **i)** *Better performance*. Through exploiting cross-task relatedness, VIENNA outperforms independent task training. **ii)** *Reduced model-size*. Training four tasks together using a single VIENNA achieves about four times model size compression, compared with training them individually. **iii)** *Improved generalization*. VIENNA performs robust on unseen environments, through learning task-shared, general representations. **iv)** *More is better*. The above conclusions are typically true when we train VIENNA on more navigation tasks. **v)** *Multisensory integration does matter*. Both visual (RGB and depth) and aural information are crucial building blocks for general-purpose navigation robot creation.

## 2 Related Work

**Embodied Navigation.** As a fundamental element in building intelligent robots, navigation has long been the focus of the scientific community [17]. The availability of building-scale 3D datasets [18–21]

---

[2]Strictly speaking, as we synthesize visually and acoustically realistic environments, the classic *image-goal nav.*, *object-goal nav.*, and *vision-language nav.* tasks in our VXN dataset are extended as *image-goal visual-audio nav.*, *visual-audio object-goal nav.*, and *visual-audio-language nav.*, respectively.

and high-performance simulation platforms [15, 22–24] led to a plethora of reproducible research of navigation in large-scale, visually-rich environments. Depending on how to specify the target goal, diverse navigation tasks are proposed to let an agent **i)** navigate to target coordinates (*point-goal nav.* [15]), **ii)** find an instance of a given object category (*object-goal nav.* [7]), **iii)** search for target photos (*image-goal nav.* [5]), **iv)** locate sound sources (*audio-goal nav.* [6]), or **v)** follow navigation instructions (*vision-language nav.* [8, 25]). Aside from these battlefields, there are some more complicated embodied tasks, such as *embodied question answering* [26], *vision-dialog nav.* [27–29], and *multiagent nav.* [30]. The community also made great strides in improving reinforcement learning (RL) algorithms capable of fulfilling specific navigation tasks, by using, for example, recurrent neural networks [8, 15, 31], map building [3, 32–41], path planning [42–45], cross-modal attention [41, 46–49], synthesized or unlabeled data [50–55], and external knowledge [7, 56]. However, though the learning algorithm is general – RL, each solution is not; each navigation agent can only handle the one task it was trained on.

With various navigation tasks and task-specific navigation solutions, a critical question arises *whether we can build a single general agent that works well for multiple navigation tasks*. In response, we make two unique contributions. First, we build a large-scale 3D dataset that supports four representative navigation tasks in continuous and realistic environments. In contrast, prior navigation datasets are built upon different platforms and with certain assumptions/configurations (*e.g.*, sparse navigation graphs [8], discrete world representation [6]), making them hard to explore different navigation tasks in unified and standardized environments. Second, we create a generalist agent which is capable of undertaking a set of navigation tasks of different modalities/domains, and is equipped with multimodal sensors (*i.e.*, RGB, depth, audio) to better address real-world scenarios. However, existing navigation agents are trained one task at the time, each new task requiring to train a new agent instance.

**Multitask Learning (MTL).** MTL [12], inspired by the human ability to transfer knowledge across different tasks [57], has led to wide success in computer vision [58–60] and natural language processing [61]. Related efforts were made along three directions [13]: **i)** *architecture design* (*i.e.*, how to partition the model into task-specific and shared components) [62–65], **ii)** *optimization* (*i.e.*, how to balance learning between different tasks) [66–71], and **iii)** *task relationship learning* (*i.e.*, how to learn and utilize task relationships to improve learning) [72–74]. In the field of MTRL [75–81], recent solutions explored knowledge transfer [82], modular networks [83, 84], and policy distillation [85, 86]. A few robotics benchmarks [87–89] are also proposed for MTRL. However, most of these efforts were based upon low-dimension observations, *e.g.*, grid-world like or game environments. To the best of our knowledge, there are two prior work [48, 90] that addressed multitask navigation, but they only consider two closely-related, language-guided navigation tasks with the same input modalities.

Drawing inspiration from these efforts, we seek for a "universal" agent that can complete multiple navigation tasks with a single agent instance, and distinguish ourselves by **i)** joint learning of four navigation tasks with diverse input modalities, **ii)** visually complex and acoustically realistic operation space, **iii)** multisensory integration, and **iv)** fully-attentive architecture based parse-and-query regime.

**Auxiliary Learning in Embodied Navigation.** There are a group of algorithms that exploit complementary objectives from auxiliary tasks to facilitate navigation policy learning. Specifically, *supervised auxiliary tasks* expose privileged information to the agent (*e.g.*, depth [91], surface normals [92], semantics [26], *etc*.). *Self-supervised auxiliary tasks* derive free supervisory signals from the agent's own experience (*e.g.*, next-step visual feature prediction [93], predictive modeling [94], loop closure prediction [91], temporal distance estimation [95], navigation progress estimation [96, 97], *etc*.).

Although auxiliary learning based navigation models are also trained on a set of tasks, their ideas are far away from ours. These models still focus on only a single "main" navigation task with extra aid of auxiliary intermediate objectives, while we aim to capture and utilize common knowledge of a collection of different navigation tasks to enhance the performance on all the tasks. Moreover, their auxiliary tasks, in principle, can be utilized by our agent, but they cannot handle our task setting.

**Transformer in Embodied Navigation and MTL.** Inspired by the great success of Transformer [16] in sequence transduction tasks, a few recent methods applied Transformer for certain navigation tasks [40, 98–102]. Rather than sharing similar advantages in long-term memory and cross-modal information fusion, our method further formulates different navigation tasks as a unified process of active goal parsing and supportive information query. Through cleverly encoding all task-specific embeddings into goal parsing, our agent is able to explicitly leverage cross-task knowledge to boost different navigation tasks. There are also a few notable studies that exploit Transformer-like network architectures for MTL [103–106], while none of them addresses embodied visual tasks.

Table 1: Data splits and the number of navigation episodes in our `VXN` dataset (§3.2).

| Navigation Task | train (58 environments) | val seen (58 environments) | val unseen (11 environments) |
|---|---|---|---|
| *Audio-goal* | 2.0M episodes | 500 episodes | 500 episodes |
| *Vision-language* | 10,819 episodes | 778 episodes | 1,839 episodes |
| *Object-goal* | 2.6M episodes | 500 episodes | 2,195 episodes |
| *Image-goal* | 5.0M episodes | 495 episodes | 495 episodes |
| Total | 9.6M episodes | 2,273 episodes | 5,029 episodes |

Table 2: Comparison (§3.2) of navigation datasets (MT: multitask; CS: continuous space; VR/AR: visual/audio realistic; PA: panorama).

| Navigation Dataset | Year | MT | CS | VR | AR | PA |
|---|---|---|---|---|---|---|
| EQA [26] | 2018 | | | | | |
| Habitat-PointGoal [15] | 2019 | | ✓ | ✓ | | |
| R2R [8] | 2018 | | | ✓ | | |
| VLN-CE [25] | 2020 | | ✓ | ✓ | | |
| Gibson-ImageGoal [3] | 2020 | | ✓ | ✓ | | ✓ |
| SoundSpaces [6] | 2020 | | | ✓ | ✓ | |
| VXN | 2022 | ✓ | ✓ | ✓ | ✓ | ✓ |

**Pretraining in Embodied Navigation.** A series of methods decompose embodied tasks into visual (and linguistic) representation learning and policy training [53, 92, 107–109]. They pretrain a general model on easily-acquired viual or multimodal data (*e.g.*, image captions) and fine-tune the policy for "downstream" navigation tasks. Though showing improved generalization and transfer abilities, the result is still a collection of independent task-specific models rather than a single agent instance.

# 3 `VXN` Dataset for Multitask Multimodal Embodied Navigation

## 3.1 Task Collection and Dataset Acquisition

`VXN` includes four famous navigation tasks, *i.e.*, *image-goal nav.* [5], *audio-goal nav.* [6], *object-goal nav.* [7], and *vision-language nav.* [8]. These tasks are with different input modalities/domains (*i.e.*, visual, audio, semantic tag, and language); their original datasets adopt different world representations (*i.e.*, graph based [8] *vs* discrete [6] *vs* continuous [15]), environment configurations (*i.e.*, visually poor [22] *vs* perception rich [8] *vs* audiovisual realistic [6]), and success criteria (*i.e.*, 3 m [8] *vs* 1 m [5] *vs* 1 m [7] *vs* 1 m [100]). Hence, to study these four tasks in a single learning system, it is desired to build a standardized dataset that initiates them with similar problem settings, *e.g.*, dynamic transition, world representations, and audiovisual properties, instead of simply combining several single-task navigation datasets together. On the other hand, it is wise to maximize the reuse of existing datasets, ensuring continuity and compatibility w.r.t. former research, and reducing data annotation cost.

As many previous navigation datasets [6, 8, 110] are built upon Matterport3D (MP3D) [19] environments and Habitat [15] simulator, we derive a unified, multitask navigation dataset – `VXN` – by converting previous *task-specific* datasets to *standardized, continuous, audiovisual-rich environments*:

- Our ***audio-goal nav.*** is built upon SoundSpaces [6], which offers audio renderings for MP3D and allows to navigate sounding targets, or conduct point navigation with extra aid of audio cues. Due to heavy acoustic simulation cost, [6] uses a grid-based world model: it samples room impulse res- ponse over a discrete, horizontal plane (1.5 m above the floor with 0.5 m × 0.5 m grid size). We devise an audio simulator to efficiently transfer grid-level audio renderings into continuous setting (*cf.* §3.2).
- Our ***vision-language nav.*** is built upon R2R [8], which labels MP3D with linguistic navigation ins- tructions. R2R is yet bounded to graph-based world representation – each scene can be only observed from a few fixed points (∼117) and environment topologies are pre-given. We use [25] to convert R2R to the continuous setting, and then adopt [6] and our audio simulator for audio rendering.
- Our ***image-goal nav.*** is built upon Habitat [15] ImageNav repository [111], which is for photo target guided navigation in MP3D environments. Again, continuous audio rendering is made.
- Our ***object-goal nav.*** is built upon Habitat2020 ObjectNav challenge [110], which requires an agent to navigate a MP3D environment to find an instance of an object class. A total of 21 visually well defined object categories (*e.g.*, *chair*) are considered and audio rendering is also made; but the GPS + Compass sensor, used in [110], is not adopted in `VXN`, for formalizing different task settings.

## 3.2 Task Setting and Dataset Design

**Panoramic Visual Simulator.** With Habitat API, 360° egocentric RGBD view is rendered at 300 fps.

**Audio Simulator.** With [6], ambisonics are generated at locations sampled in MP3D scenes and converted to binaural audio [112], *i.e.*, an agent emulates two human-like ears. To synthesize continuous auditory scenes, we use [113] for real-time binaural room impulse responses (BRIRs) interpolation. We adopt Dynamic Time Wrapping [114] to temporally align left and right ear BRIRs and then map the

warped *interpolated* vectors back into the "unwarped" time domain, to get BRIRs at arbitrary locations and directions. As in [100], the sounds of the 21 object categories [110] in *object-goal nav.* are used for audio rendering. Moreover, the sounds are associated with the objects of same semantic categories to ensure generating semantically meaningful and contextual audio [100]. For *image-goal nav.*, *object-goal nav.*, and *vision-language nav.*, the audio is used as background sound, which can reveal the geometry of environment [6], complement the visual cues, and make the tasks closer to the real-world. For *audio-goal nav.*, the navigation target is directly specified by the audio.

**Episodes and Dataset Splits.** In VXN, each episode is defined as a tuple: ⟨scene, audio waveform, agent start location, agent start rotation, goal location, target description⟩. We use the standard 58/11/18 `train`/`val`/`test` split [115] of MP3D environments. Since previous navigation datasets [8, 15, 110] keep `test` annotations private, we only use `train` and `val` environments to create VXN (*cf.* Table 1).

**Action Space.** We adopt a panoramic action space, which is widely used in recent embodied robotic tasks [3, 43]. Specifically, the panoramic view is horizontally discreted into a total of 12 sub-views. Agents can move towards a sub-view 0.25 m or `stop`.

**Success Criterion.** An episode is considered as successful if the agent **i)** executes `stop` action, **ii)** within 1 m of the goal location, and **iii)** within a time horizon of 500 actions (as in [15, 92, 116]).

**Dataset Features.** As shown in Table 2, VXN poses greater challenges: the agent needs to master four navigation tasks with various input modalities in continuous, audiovisual complex environments, mine cross-task knowledge, and reason intelligently about all the senses available to it (RGB, depth, audio).

# 4 Our Approach

**Problem Statement.** In *single-task navigation*, an agent learns to reach a goal position. This is typically formulated in a RL framework that solves a partially observable Markov decision process [117]: a tuple $(\mathcal{S}, \mathcal{A}, \mathcal{G}, O, P, R, \gamma)$, where $\mathcal{S}$, $\mathcal{A}$, $\mathcal{G}$ are sets of *states*, *actions* and *targets*, $o_t = O(s_t)$ denotes the local observation at global state $s_t \in \mathcal{S}$ at epoch (decision step) $t$, $P(s_{t+1}|s_t, a_t)$ is the transition probability from $s_t$ to $s_{t+1}$ given action $a_t \in \mathcal{A}$, $R(s, a) \in \mathbb{R}$ gives the *reward*, and $\gamma \in (0, 1)$ discounts future rewards. The agent uses a *policy* $\pi(a|o, g)$ to produce its action $a$, conditioned on its local observation $o$ and target goal $g \in \mathcal{G}$, and optimizes its accumulated discounted reward $J = \sum_{t'=t}^{T} \gamma^{t'-t} R(s_{t'}, a_{t'})$.

In our *multitask navigation*, a single agent needs to master $K = 4$ tasks, *i.e.*, {*audio-goal*, *object-goal*, *image-goal*, *vision-language*} in VXN environments. We formalize this as a MTRL problem: $\{(\mathcal{S}, \mathcal{A}, \mathcal{G}_k, O, P, R_k, \gamma_k)\}_{k=1}^{K}$, where the agent concurrently learns $K$ task-specific policies $\pi_{1:K}$ that maximize the rewards $J_{1:K}$. The single multitask agent is expected to exploit cross-task knowledge to achieve close or better navigation performance on the $K$ tasks, compared with training $K$ single-task agents individually.

**Transformer Preliminary.** The core of Transformer [16] is an attention function (denoted as $f_{\text{ATT}}$), which takes a query sequence $\boldsymbol{x} \in \mathbb{R}^{n \times d}$ and a context sequence $\boldsymbol{y} \in \mathbb{R}^{m \times d}$ as inputs, and outputs:

$$\tilde{\boldsymbol{y}} = f_{\text{ATT}}(\boldsymbol{x}, \boldsymbol{y}) = \text{softmax}\big((\boldsymbol{x}\boldsymbol{W}^q)(\boldsymbol{y}\boldsymbol{W}^k)^\top / \sqrt{d}\big)(\boldsymbol{y}\boldsymbol{W}^v). \qquad (1)$$

where $\tilde{\boldsymbol{y}} \in \mathbb{R}^{n \times d}$ is with the same length $n$ and embedding dimension $d$ as $\boldsymbol{x}$, and $\boldsymbol{W}^{q,k,v} \in \mathbb{R}^{d \times d}$ are learnable *query*, *key*, and *value* projection matrices, respectively. Note that Eq. 1 is applicable to both *self-attention* in Transformer encoder (*i.e.*, $\boldsymbol{x} \equiv \boldsymbol{y}$), and *cross-attention* in Transformer decoder (*i.e.*, $\boldsymbol{x} \neq \boldsymbol{y}$). Further, each Transformer layer block can be given as:

$$\boldsymbol{x}' = \boldsymbol{x} + f_{\text{MHA}}(\boldsymbol{x}, \boldsymbol{y}) \in \mathbb{R}^{n \times d}, \qquad \boldsymbol{z} = \boldsymbol{x}' + f_{\text{MLP}}(\boldsymbol{x}') \in \mathbb{R}^{n \times d}, \qquad (2)$$

where $f_{\text{MHA}}$ refers to a *multi-head attention* layer, derived by computing several $f_{\text{ATT}}$ in parallel, and $f_{\text{MLP}}$ is multi-layer perceptron. The layer normalization is omitted for brevity.

**Core Idea.** Built upon a Transformer encoder-decoder architecture, our VIENNA unifies the four VXN tasks as an attention-based, *parse-and-query* framework: the target description $g \in \mathcal{G}_k$ is *online* parsed into a set of embeddings, which are used to "query" the encoded episode history; the retrieved supportive cues are fed into the corresponding policy $\pi_k$ for decision making. To better handle multiple tasks, VIENNA **i)** learns task-wise context and involves all the task-specific embeddings into target parsing, **ii)** shares representations among tasks, **iii)** lets task-specific policies $\pi_{1:K}$ reuse knowledge, and **iv)** trains the polices via a multitask version of Distributed Proximal Policy Optimization (DPPO) [118].

VIENNA has three modules (*cf.* Fig. 2): **i)** an *episodic encoder* (§4.1) that fuses multisensory cues and encodes the full episode history of navigation; **ii)** a *target parser* (§4.2) that actively interprets the

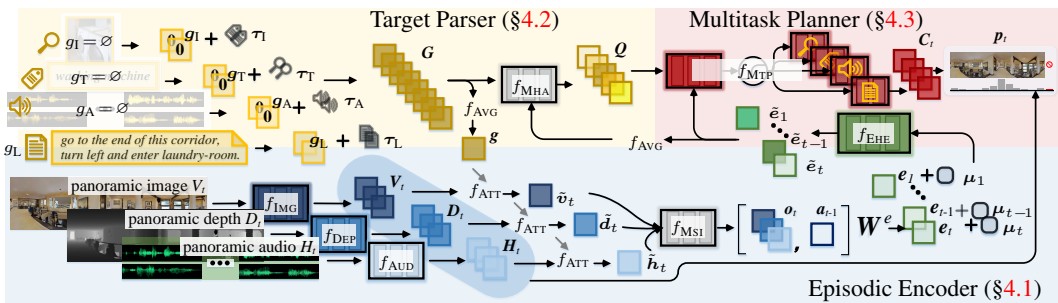

Figure 2: Detailed network architecture of VIENNA, at epoch $t$ in a *vision-language nav.* episode.

target specification into several embeddings; and **iii)** a *multitask planner* (§4.3) that uses the target embeddings to query encoded episodic history and leverages the returned context for action prediction.

## 4.1 Episodic Encoder

At the start of each episode, VIENNA receives a target description $g \in \{$goal image, target sound, target class, language instruction$\}$ and derives an embedding vector $\boldsymbol{g} \in \mathbb{R}^d$ (detailed in §4.2). At each epoch $t$, VIENNA has a $360°$ egocentric audiovisual perception $o_t$, *i.e.*, RGB+depth+audio, of its surrounding.

**Intra-Modal Encoders.** A *visual encoder* $f_{\text{IMG}}$ maps perceived panoramic image $V_t \in \mathbb{R}^{12 \times 224 \times 224 \times 3}$ into visual features $\boldsymbol{V}_t = [\boldsymbol{v}_{1,t}, \cdots, \boldsymbol{v}_{12,t}] \in \mathbb{R}^{12 \times d}$, where $\boldsymbol{v}_{i,t} \in \mathbb{R}^d$ is the feature vector of $i$-th sub-view in $V_t$. Similarly, a *depth encoder* $f_{\text{DEP}}$ and an *audio encoder* $f_{\text{AUD}}$ map the perceived panoramic depth image $D_t \in \mathbb{R}^{12 \times 256 \times 256 \times 1}$ and spectogram tensor of binaural sound (collected over 12 horizontal directions) $H_t \in \mathbb{R}^{12 \times 41 \times 44 \times 2}$ into depth and audio features, *i.e.*, $\boldsymbol{D}_t = [\boldsymbol{d}_{1,t}, \cdots, \boldsymbol{d}_{12,t}] \in \mathbb{R}^{12 \times d}$, and $\boldsymbol{A}_t = [\boldsymbol{a}_{1,t}, \cdots, \boldsymbol{a}_{12,t}] \in \mathbb{R}^{12 \times d}$, respectively.

**Target-Guided Cross-Modal Encoder.** With the target description vector $\boldsymbol{g} \in \mathbb{R}^d$, cross-attention $f_{\text{ATT}}$ (*cf.* Eq.1) is separately applied over $\boldsymbol{V}_t$, $\boldsymbol{D}_t$, and $\boldsymbol{H}_t$ to assemble target-related sensory information:

$$\tilde{\boldsymbol{v}}_t = f_{\text{ATT}}(\boldsymbol{g}, \boldsymbol{V}_t) \in \mathbb{R}^d, \quad \tilde{\boldsymbol{d}}_t = f_{\text{ATT}}(\boldsymbol{g}, \boldsymbol{D}_t) \in \mathbb{R}^d, \quad \tilde{\boldsymbol{h}}_t = f_{\text{ATT}}(\boldsymbol{g}, \boldsymbol{H}_t) \in \mathbb{R}^d. \quad (3)$$

Then $\tilde{\boldsymbol{v}}_t$, $\tilde{\boldsymbol{d}}_t$, and $\tilde{\boldsymbol{h}}_t$ are concatenated for attention based *multisensory information integration* (MSI):

$$\boldsymbol{o}_t = f_{\text{MSI}}([\tilde{\boldsymbol{v}}_t, \tilde{\boldsymbol{d}}_t, \tilde{\boldsymbol{h}}_t]) \in \mathbb{R}^{3 \times d}, \quad (4)$$

where $f_{\text{MSI}}$ is achieved by stacking two self-attention based Transformer blocks (*cf.* Eq. 2).

**Episodic History Encoder.** At epoch $t$, the multimodal observation embedding $\boldsymbol{o}_t \in \mathbb{R}^{3 \times d}$ and latest navigation action embedding $\boldsymbol{a}_{t-1} \in \mathbb{R}^d$, are together projected into a compact "*navigation token*":

$$\boldsymbol{e}_t = [\boldsymbol{o}_t, \boldsymbol{a}_{t-1}] \boldsymbol{W}^e \in \mathbb{R}^d. \quad (5)$$

All the past navigation tokens, $\boldsymbol{e}_{1:t}$, summed with corresponding epoch embedding vectors, $\boldsymbol{\mu}_{1:t} \in \mathbb{R}^d$, are collected into a sequence and fed into an *episode history encoder* (EHE) to get contextualized history representation:

$$[\tilde{\boldsymbol{e}}_1, \cdots, \tilde{\boldsymbol{e}}_t] = f_{\text{EHE}}([\boldsymbol{e}_1 + \boldsymbol{\mu}_1, \cdots, \boldsymbol{e}_t + \boldsymbol{\mu}_t]), \quad (6)$$

where $f_{\text{EHE}}$ is implemented as four self-attention based Transformer blocks (*cf.* Eq. 2). In this way, VIENNA is able to store and access its entire episode history of audiovisual observations and actions, leading to persistent memorization and long-term reasoning. The attended history representation $\tilde{\boldsymbol{e}}_{1:t}$ will serve as informative context for predicting the navigation action $a_t$ at epoch $t$ (detailed in §4.3).

## 4.2 Target Parser

VIENNA is equipped with a *target parser* that actively interprets the

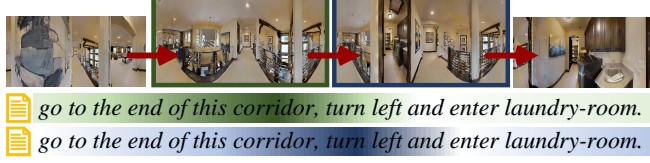

*go to the end of this corridor, turn left and enter laundry-room.*
*go to the end of this corridor, turn left and enter laundry-room.*

Figure 3: Attention visualization of online target parsing (Eq. 8).

target $g$ (no matter it is specified as a photo $g_I$, sound $g_A$, semantic tag $g_T$, or linguistic instruction $g_L$) as a group of target embeddings, conditioned on the progress of the navigation episode. Guided by the online created target embeddings, valuable context are selected from episodic experiences $\tilde{e}_{1:t}$ for flexible decision-making. VIENNA thus formulates various navigation tasks in a unified scheme, allowing to exploit cross-task knowledge.

In *image-goal nav.*, a goal image $g_I \in \mathbb{R}^{224 \times 224 \times 3}$ is given and embedded as $\boldsymbol{g}_I = f_{\text{IMG}}(g_I) \in \mathbb{R}^{N_I \times d}$. In *audio-goal nav.*, the target is signaled by the binaural sound, *i.e.*, $g_A = H_t \in \mathbb{R}^{12 \times 41 \times 44 \times 2}$ and $\boldsymbol{g}_A = \boldsymbol{H}_t = f_{\text{AUD}}(H_t) \in \mathbb{R}^{N_A \times d}$. In *object-goal nav.*, the target is specified by a semantic tag $g_T \in \{\text{table, bed}, \cdots\}$, and embedded into a class vector $\boldsymbol{g}_T \in \mathbb{R}^{1 \times d}$. In *vision-language nav.*, a language-based trajectory instruction $g_L$ is given and mapped into a sequence of word vectors $\boldsymbol{g}_L \in \mathbb{R}^{N_L \times d}$ by a bi-LSTM. At the start of each episode, we first build an *augmented target description embedding* $\boldsymbol{G} \in \mathbb{R}^{4N_G \times d}$:

$$\boldsymbol{G} = \left[ \boldsymbol{g}_I' + [\boldsymbol{\tau}_I]^{N_G}, \ \boldsymbol{g}_A' + [\boldsymbol{\tau}_A]^{N_G}, \ \boldsymbol{g}_T' + [\boldsymbol{\tau}_T]^{N_G}, \ \boldsymbol{g}_L' + [\boldsymbol{\tau}_L]^{N_G} \right], \tag{7}$$

where $N_G = \max(N_I, N_A, 1, N_L)$, $\boldsymbol{\tau}_{I,A,T,L} \in \mathbb{R}^d$ are learnable task embedding vectors, and $[\cdot]^{N_G}$ copies its input $N_G$ times. Assuming VIENNA is in an *image-goal nav.* episode, we have $\boldsymbol{g}_I = f_{\text{IMG}}(g_I) \in \mathbb{R}^{N_I \times d}$, and $\boldsymbol{g}_A = [\boldsymbol{0}]^{N_A \times d}$, $\boldsymbol{g}_T = [\boldsymbol{0}]^{1 \times d}$, $\boldsymbol{g}_L = [\boldsymbol{0}]^{N_L \times d}$. We pad $\boldsymbol{g}_{I,A,T,L}$ to a unified length $N_G$, by replication, so as to get $\boldsymbol{g}_{I,A,T,L}'$ and make them contribute equally to $\boldsymbol{G}$. We collect all the task-type embeddings $\boldsymbol{\tau}_{I,A,T,L}$ and current target description $g \in \{g_I, g_A, g_T, g_L\}$ into $\boldsymbol{G}$. In §4.3, we will show this strategy is essential for making use of cross-task knowledge. The target description vector $\boldsymbol{g} \in \mathbb{R}^d$ used in Eq. 3 is given as: $\boldsymbol{g} = f_{\text{AVG}}(\boldsymbol{G})$, where $f_{\text{AVG}}$ stands for the average pooling operation.

At epoch $t$, the target parser comprehends the augmented target description embedding $\boldsymbol{G}$ as a set of $N$ compact embeddings on-the-fly, conditioned on its episodic, contextualized history encoding $\tilde{e}_{1:t}$:

$$\boldsymbol{Q}_t = [\boldsymbol{q}_t^1, \cdots, \boldsymbol{q}_t^N] = f_{\text{MHA}}(f_{\text{AVG}}(\tilde{\boldsymbol{e}}_{1:t}), \boldsymbol{G}) \in \mathbb{R}^{N \times d}, \tag{8}$$

where $f_{\text{MHA}}$ is a $N$-head attention layer (*cf.* Eq. 2), *i.e.*, explain $\boldsymbol{G}$ in different ways, with consideration of current episodic navigation progress. Each of the target embedding vectors $\boldsymbol{q}_t$ can be viewed as a specific, time-varying goal, used to guide action selection $a_t$ at epoch $t$. As shown in Fig. 3, given a navigation instruction "*go to the end of this corridor, turn left and* $\cdots$", the agent focuses more on "*go to the end of this corridor*" at the start of the navigation episode. After reaching the end of the corridor, the agent shifts its attention to "*turn left*". Here a collection of $N$ target embeddings $\boldsymbol{q}^{1:N}$ are generated at each epoch $t$, allowing the agent to capture different aspects of target-related information and making the time-varying goal well-planned. For instance, there may exist several essential landmarks in a goal image, or multiple discriminative audio clips in target-emitted sound; during navigation, the agent should be able to pay attention to all these informative clues simultaneously.

## 4.3 Multitask Planner

At epoch $t$, a multitask planner (MTP) uses the diversified target embeddings $\boldsymbol{Q}_t = [\boldsymbol{q}_t^1, \cdots, \boldsymbol{q}_t^N]$ to query the episodic history $\tilde{e}_{1:t}$:

$$\boldsymbol{C}_t = f_{\text{MTP}}([\boldsymbol{q}_t^1, \cdots, \boldsymbol{q}_t^N], \ [\tilde{e}_1, \cdots, \tilde{e}_t]), \tag{9}$$

where $f_{\text{MTP}}$ is achieved by a four-layer Transformer decoder; the first two layers are shared among the four navigation tasks for capturing task-shared policies, while the last two layers are private for each task for task-specific policy learning. We empirically find such *shared trunk* based MTP design yields better performance than learning task-specific policies individually or just training one single "universal" policy (*cf.* §5.2).

The decision-making is conditioned on the retrieved context $\boldsymbol{C}_t \in \mathbb{R}^{N \times d}$, and the presentations of current multi-modal observations (*cf.* §3.2), including $\boldsymbol{V}_t = [\boldsymbol{v}_{1,t}, \cdots, \boldsymbol{v}_{12,t}] \in \mathbb{R}^{12 \times d}$, $\boldsymbol{D}_t = [\boldsymbol{d}_{1,t}, \cdots, \boldsymbol{d}_{12,t}] \in \mathbb{R}^{12 \times d}$, and $\boldsymbol{A}_t = [\boldsymbol{a}_{1,t}, \cdots, \boldsymbol{a}_{12,t}] \in \mathbb{R}^{12 \times d}$. Specifically, at epoch $t$, VIENNA makes navigate decision by choosing between the 12 current sub-views, as well as an extra STOP action. Given 12 subview action embeddings $\{\boldsymbol{b}_{i,t} \in \mathbb{R}^{3 \times d}\}_{i=1}^{12}$, *i.e.*, $\boldsymbol{o}_{i,t} = [\boldsymbol{v}_{i,t}, \boldsymbol{d}_{i,t}, \boldsymbol{h}_{i,t}]$ as well as a STOP action embedding, *i.e.*, $\boldsymbol{b}_{13,t} = \vec{0}$, represented as an all-zero vector, VIENNA predicts a probability distribution $\boldsymbol{p}_t = [p_{1,t}, \cdots, p_{13,t}]$:

$$p_{i,t} = \text{softmax}_i(f_{\text{AVG}}(\boldsymbol{C}_t)\boldsymbol{W}^p \boldsymbol{b}_{i,t}) \in [0,1], \qquad \text{where } i \in \{1, \cdots, 13\}. \tag{10}$$

As the task embeddings $\boldsymbol{\tau}_{I,A,T,L}$ are encoded into $\boldsymbol{Q}_t$, which is used to find supportive cues from episodic observations $\tilde{\boldsymbol{e}}_{1:t}$ for long-term reasoning and decision-making, $\boldsymbol{\tau}_{I,A,T,L}$ are essentially

trained as task-wise context – they are sensitive to task-related cues. Thus collecting $\tau_{I,A,T,L}$ into $G$ (*cf.* Eq. 7) enables a clever use of cross-task knowledge. For instance, during *image-goal nav.*, $\tau_A$ can help the agent notice some informative audio signals, $\tau_T$ can alert the agent to visually essential semantics, while $\tau_I$ can be activated by crucial landmarks. Related experiments can be found in §5.2.

### 4.4 MTRL based Multitask Navigation Training

**Reward Design.** With standardized `VXN` environments, VIENNA adopts a same reward function for the four navigation tasks, *i.e.*, $R_1 = \cdots = R_4$. Concretely, $R_{1:4}$ has four terms, *i.e.*, a sparse success reward $r_{\text{success}}$, a progress reward $r_{\text{progress}}$, a slack reward $r_{\text{slack}}$, and an exploration reward $r_{\text{explore}}$. $r_{\text{success}} = 2.5$ is only received at the end of a successful episode. $r_{\text{progress}} = -\Delta_{\text{geo\_dist}}$ offers dense signals indicating the progress that an action contributes: $\Delta_{\text{geo\_dist}}$ gives the change in geodesic distance to the goal position by performing the action. $r_{\text{slack}} = -10^{-3}$, received at each epoch, penalizes redundant actions. $r_{\text{explore}}$ [119] divides each environment into a voxel grid with $2.5\,\text{m} \times 2.5\,\text{m} \times 2.5\,\text{m}$ voxels and rewards the agent for visiting each voxel. $r_{\text{explore}}$ is defined as $0.25\eta$, where $\eta = \delta^t/\nu$ is a coefficient that decays as episode epoch $t$ and visited voxel number $\nu$ increase, and $\delta$ is a decay constant of $0.995$.

**Multitask Distributed Proximal Policy Optimization.** We present a multitask distributed proximal policy optimization (MDPPO) algorithm, which utilizes the power of parallel processing to train MTRL agents in our continuous and large-scale environments. MDPPO is built upon (DPPO) [118], a distributed version of proximal policy optimization (PPO) [120] that bounds parameter updates to a trust region to ensure stability, and distributes the computation over many parallel instances of agent and environment. Similarly, MDPPO has a server-client structure: each client worker has several agent copies that collect experiences from `VXN` environments, compute and send PPO's gradients to the server; the server worker averages the received gradients, updates the agent, and synchronizes the updated weights with the clients. For balanced multitask learning, *i.e.*, training data in `VXN` are biased between *vision-language nav.* and other navigation tasks: 10.8K *vs* 2.0∼5.0M episodes (*cf.* Table 1), each client worker is required to build four agent copies corresponding to the four `VXN` tasks.

### 4.5 Implementation Detail

**Network Architecture.** The *visual encoder* $f_{\text{IMG}}$ is made as an ImageNet [121]-pretrained ResNet50 [122]. The CNN features are fed into a linear layer for dimension compression and flattened into a feature sequence. Similarly, the *depth encoder* $f_{\text{DEP}}$ is a modified ResNet50 CNN. The *audio encoder* $f_{\text{AUD}}$, following [6], is a CNN of conv $8 \times 8$, conv $4 \times 4$, conv $3 \times 3$ and a linear layer, interleaved with ReLU. All the sensory features are combined with orientation embeddings. For the *epoch embedding* $\mu$, we use sinusoidal encoding. For the *target parser*, $N = 5$ target embeddings are generated at each epoch $t$. We set other hyper-parameters as: $d = 512$, $N_I = 16$, $N_L = 120$, $N_G = 120$.

**Training and Test.** VIENNA is trained on 32 RTX 2080 GPUs for 180 M frames, costing $4,608$ GPU hours. As in [25], we select the checkpoint for evaluation with the best SR on `val unseen`. For MDPPO, we use four client workers and set the discounted factor $\gamma$ as 0.99. We use AdamW [123] optimizer with a learning rate of $2.5 \times 10^{-4}$. Casual attention [16] is adopted to prevent the prediction at epoch $t$ from the influence of future tokens after $t$. Once trained, a single instance of VIENNA can conduct the four navigation tasks. As normal, greedy prediction is adopted for action selection.

## 5 Experiment

In §5.1, we first report comparison results for the four `VXN` tasks. In §5.2, we conduct diagnostic studies to examine the efficacy of our core model design. More results are put in the ***supplementary***.

**Baseline.** We test several open-source task-specific navigation methods [2, 5, 6, 25]. Note that [2, 5, 6] are re-trained on `VXN`, since they use different training data [5], world representation [6] (discrete *vs* continuous), or object categories [2] (6 *vs* 21). For [25], we use its check-point but the success criteria are different (3 m *vs* 1 m). Thus their scores on `VXN` are different from the original ones. In addition, we consider a `Seq2Seq` agent, which also serves as a standard baseline in [8, 25]: an LSTM planner encodes the episode history and predicts navigation actions in a sequential menner. For all the four tasks, we provide the performance of both the single-task and multitask versions of our VIENNA and `Seq2Seq`. Further, `Random policy`, *i.e.*, choosing actions randomly, is included.

Table 3: Quantitative comparison results (§5.1) on VXN dataset (ST: Single-task; MT: Multitask).

| Models | val seen | | | | val unseen | | | |
|---|---|---|---|---|---|---|---|---|
| | SR$^\uparrow$ | NE$^\downarrow$ | OR$^\uparrow$ | SPL$^\uparrow$ | SR$^\uparrow$ | NE$^\downarrow$ | OR$^\uparrow$ | SPL$^\uparrow$ |
| Random | 1.2 | 14.20 | 1.9 | 1.2 | 1.4 | 14.14 | 2.2 | 1.4 |
| Seq2Seq$_{ST}$ | 15.1 | 10.44 | 19.1 | 12.6 | 9.3 | 12.02 | 13.9 | 7.4 |
| Seq2Seq$_{MT}$ | 15.8 | 10.21 | 21.3 | 13.0 | 10.2 | 10.22 | 15.4 | 8.5 |
| Zhu *et al.* [5] | 17.7 | 9.67 | 22.0 | 13.1 | 12.0 | 10.19 | 16.6 | 8.9 |
| VIENNA$_{ST}$ | 19.9 | 9.52 | 23.2 | 13.4 | 12.6 | 9.83 | 17.1 | 9.5 |
| **VIENNA$_{MT}$** | 22.1 | 9.43 | 24.2 | 14.1 | 14.3 | 9.66 | 18.5 | 11.1 |

(a) *image-goal nav.* (IGN)

| Models | val seen | | | | val unseen | | | |
|---|---|---|---|---|---|---|---|---|
| | SR$^\uparrow$ | NE$^\downarrow$ | OR$^\uparrow$ | SPL$^\uparrow$ | SR$^\uparrow$ | NE$^\downarrow$ | OR$^\uparrow$ | SPL$^\uparrow$ |
| Random | 0.0 | 17.13 | 0.0 | 0.0 | 0.0 | 16.84 | 0.0 | 0.0 |
| Seq2Seq$_{ST}$ | 17.4 | 10.11 | 19.0 | 15.8 | 11.0 | 10.83 | 13.3 | 8.8 |
| Seq2Seq$_{MT}$ | 18.1 | 9.69 | 20.3 | 16.0 | 11.8 | 10.76 | 14.1 | 9.3 |
| Chen *et al.* [6] | 20.1 | 8.84 | 21.5 | 17.1 | 13.1 | 9.26 | 15.7 | 10.4 |
| VIENNA$_{ST}$ | 22.4 | 8.76 | 22.4 | 17.3 | 14.3 | 9.22 | 16.5 | 10.6 |
| **VIENNA$_{MT}$** | 25.3 | 8.61 | 23.9 | 17.8 | 18.7 | 8.93 | 17.9 | 12.5 |

(b) *audio-goal nav.* (AGN)

| Models | val seen | | | | val unseen | | | |
|---|---|---|---|---|---|---|---|---|
| | SR$^\uparrow$ | NE$^\downarrow$ | OR$^\uparrow$ | SPL$^\uparrow$ | SR$^\uparrow$ | NE$^\downarrow$ | OR$^\uparrow$ | SPL$^\uparrow$ |
| Random | 0.8 | 7.67 | 1.0 | 0.8 | 2.0 | 7.56 | 2.1 | 1.7 |
| Seq2Seq$_{ST}$ | 26.7 | 6.61 | 33.3 | 14.4 | 8.9 | 7.31 | 11.1 | 4.4 |
| Seq2Seq$_{MT}$ | 28.7 | 6.45 | 35.0 | 15.8 | 10.8 | 7.13 | 14.0 | 4.8 |
| Chaplot *et al.* [2] | 31.3 | 6.15 | 35.2 | 16.7 | 17.6 | 7.08 | 21.3 | 7.5 |
| VIENNA$_{ST}$ | 33.2 | 6.11 | 36.4 | 17.1 | 18.5 | 6.95 | 22.1 | 8.1 |
| **VIENNA$_{MT}$** | 33.3 | 5.92 | 37.8 | 17.7 | 19.4 | 6.77 | 25.1 | 10.7 |

(c) *object-goal nav.* (OGN)

| Models | val seen | | | | val unseen | | | |
|---|---|---|---|---|---|---|---|---|
| | SR$^\uparrow$ | NE$^\downarrow$ | OR$^\uparrow$ | SPL$^\uparrow$ | SR$^\uparrow$ | NE$^\downarrow$ | OR$^\uparrow$ | SPL$^\uparrow$ |
| Random | 0.0 | 8.89 | 0.0 | 0.0 | 0.0 | 8.92 | 0.0 | 0.0 |
| Seq2Seq$_{ST}$ | 13.2 | 7.54 | 17.7 | 12.1 | 5.2 | 8.49 | 9.7 | 4.6 |
| Seq2Seq$_{MT}$ | 17.6 | 7.29 | 22.8 | 15.4 | 7.6 | 8.21 | 13.4 | 6.5 |
| Krantz *et al.* [25] | 23.7 | 7.22 | 25.9 | 21.2 | 11.0 | 7.60 | 16.2 | 10.2 |
| VIENNA$_{ST}$ | 23.9 | 7.16 | 26.1 | 22.2 | 14.3 | 7.35 | 18.5 | 12.5 |
| **VIENNA$_{MT}$** | 26.5 | 7.08 | 27.9 | 24.1 | 16.3 | 7.26 | 20.6 | 15.7 |

(d) *vision-language nav.* (VLN)

(a) *image-goal nav.*    (b) *audio-goal nav.*    (c) *object-goal nav.*    (d) *vision-language nav.*

Figure 4: Training curves of VIENNA agents compared to Seq2Seq agents on the four VXN tasks (§5.1).

**Metric.** Four widely-used metrics are adopted for evaluation: i) *Success Rate* (**SR**); ii) *Navigation Error* (NE); iii) *Oracle success Rate* (OR); and iv) *Success rate weighted by Path Length* (SPL) [115].

## 5.1 Performance Benchmarking

Table 3 reports the comparison results on the four VXN tasks. Some key conclusions are list below:

- VIENNA obtains impressive results, under `val seen` and `unseen` sets, across all the tasks and evaluation metrics. This proves the versatility of VIENNA and the power of our parse-and-query regime.
- VIENNA consistently outperforms Seq2Seq, no matter they are trained on single tasks individually or multiple tasks jointly. Compared with other task-specific competitors [2,5,6,25], VIENNA gains comparable results on *audio-goal nav.* and *object-goal nav.*, and performs better on *image-goal nav.* and *vision-language nav.* tasks. These results verify the effectiveness of our model design.
- When considering the performance gain from the single-task setting to multitask, VIENNA yields more promising results, compared with Seq2Seq. For example, in Table 3a, VIENNA$_{MT}$ outperforms VIENNA$_{ST}$ by 2.2% SR and 1.7% SR, on `val seen` and `unseen`, respectively; however, in the same condition, Seq2Seq$_{MT}$ only provides 0.7% and 0.9% SR gains over Seq2Seq$_{ST}$. These results demonstrates that VIENNA can make a better use of cross-task knowledge.
- When considering the performance gap between seen and unseen environments, VIENNA$_{MT}$ is more favored than its single-task counterpart, VIENNA$_{ST}$. For instance, in Table 3c, VIENNA$_{ST}$ suffers from relatively large performance drop, *i.e.*, 33.2%→18.5% SR; however, VIENNA$_{MT}$ shows reduced degradation, *i.e.*, 33.3%→19.4% SR, in unseen environments. This indicates that investigating inter-task relatedness may help to strengthen the generalizability of navigation agents.
- The above results are particularly impressive considering the advantage of VIENNA in efficient parameter utilization, *i.e.*, VIENNA$_{MT}$ (31 M) *vs* VIENNA$_{ST}$× 4 (101 M) *vs* Seq2Seq$_{MT}$ (27 M) *vs* Seq2Seq$_{ST}$× 4 (93 M) *vs* [5] + [6] + [2] + [25] (165M = 40 M + 45 M + 38 M + 42 M).

Fig. 4 plots the training curves of VIENNA$_{ST/MT}$ compared to Seq2Seq$_{ST/MT}$ for the four VXN tasks in `unseen` envs. Aligning with the results in Table 3, VIENNA outperforms Seq2Seq, and benefits more from multiple task learning. This shows that VIENNA makes a better use of cross-task knowledge.

Table 4: Ablation studies (§5.2) with *audio-goal nav.* (AGN) and *vision-language nav.* (VLN) tasks.

| Modality (§4.2) | AGN (SR$^\uparrow$) seen/unseen | VLN (SR$^\uparrow$) seen/unseen |
|---|---|---|
| RGB *only* | 2.5 / 2.1 | 21.1 / 11.2 |
| audio *only* | 23.1 / 15.9 | 0.3 / 0.2 |
| RGBD *only* | 2.4 / 2.3 | 21.9 / 12.5 |
| RGBD+audio | 25.3 / 18.7 | 26.5 / 16.3 |

(a) multisensory integration

| $G$ (Eq. 7) | AGN (SR$^\uparrow$) seen/unseen | VLN (SR$^\uparrow$) seen/unseen |
|---|---|---|
| episodic target only | 23.1 / 17.2 | 22.7 / 14.1 |
| episodic target + episodic task embedding | 24.2 / 17.9 | 25.1 / 15.5 |
| augmented target des. embed. | 25.3 / 18.7 | 26.5 / 16.3 |

(b) augmented target description embedding

| $Q_t$ (Eq. 8) | AGN (SR$^\uparrow$) seen/unseen | VLN (SR$^\uparrow$) seen/unseen |
|---|---|---|
| $N=1$ | 21.0 / 15.6 | 22.2 / 13.2 |
| $N=3$ | 23.8 / 17.8 | 25.7 / 15.5 |
| $N=5$ | 25.3 / 18.7 | 26.5 / 16.3 |
| $N=7$ | 24.9 / 18.3 | 26.2 / 16.2 |

(c) diversified target parsing

| $F_{\text{MTP}}$ (Eq. 9) | AGN (SR$^\uparrow$) seen/unseen | VLN (SR$^\uparrow$) seen/unseen |
|---|---|---|
| separate | 23.2 / 17.4 | 25.0 / 15.1 |
| 1-shared | 24.4 / 18.1 | 25.8 / 15.7 |
| 2-shared | 25.3 / 18.7 | 26.5 / 16.3 |
| 3-shared | 24.8 / 17.9 | 25.9 / 15.5 |
| *all*-shared | 24.1 / 17.6 | 25.6 / 15.3 |

(d) multitask planner

| $R$ (§4.4) | AGN (SR$^\uparrow$) seen/unseen | VLN (SR$^\uparrow$) seen/unseen |
|---|---|---|
| $r_{\text{success}}$ | 2.1 / 1.5 | 5.5 / 2.3 |
| $r_{\text{success}}+r_{\text{progress}}$ | 22.5 / 16.3 | 23.8 / 14.7 |
| $r_{\text{success}}+r_{\text{progress}}+r_{\text{slack}}$ | 23.1 / 16.9 | 24.3 / 15.1 |
| $r_{\text{success}}+r_{\text{slack}}+r_{\text{progress}}+r_{\text{explore}}$ | 25.3 / 18.7 | 26.5 / 16.3 |

(e) reward function

| Task | AGN (SR$^\uparrow$) seen/unseen | VLN (SR$^\uparrow$) seen/unseen |
|---|---|---|
| single task | 22.1 / 15.4 | 23.8 / 14.1 |
| AGN+VLN | 22.7 / 16.1 | 24.4 / 14.9 |
| AGN+VLN+IGN | 24.1 / 17.3 | 25.1 / 15.5 |
| AGN+VLN+ IGN+OGN | 25.3 / 18.7 | 26.5 / 16.3 |

(f) multitask learning

## 5.2 Ablative Study

To thoroughly test the efficacy of crucial components of VIENNA, we conduct a series of diagnostic studies on *vision-language nav.* and *audio-goal nav.* tasks. The results are summarized in Table 4.

**Multisensory Integration.** Agents in VXN are equipped with a multimodal sensor so as to find the target by both seeing and hearing and make our navigation setting closer the real-world. We first study the influence of different sensory signals (*i.e.*, RGB, depth, audio) by training VIENNA with varying sensory modalities. As shown in Table 4a, fusing multimodal sensory cues (*i.e.*, RGBD + audio) is more favored. For example, in VLN, although considering audio alone brings poor performance, supplementing RGBD perception with audio yields notable improvements. This suggests audio is complementary to visual sensory in capturing physical and semantic properties of environments.

**Augmented Target Description Embedding.** To better master cross-task knowledge, VIENNA augments its episodic targets with all the four learnable task embeddings $\tau_{I,A,T,L}$ (*cf.* Eq. 7). We compare this design against two variants in Table 4b, and find such a strategy is conducive to the performance. This is because, through end-to-end training, $\tau_{I,A,T,L}$ carry task-specific knowledge, *e.g.*, $\tau_A$ is associated with some discriminative audio clips; $\tau_I$ focuses on essential visual landmarks. By taking $\tau_{I,A,T,L}$ together, VIENNA use key knowledge of different tasks in single task episodes.

**Diversified Target Parsing.** We online parse the augmented target description embedding $G$ into $N$ target embeddings $Q_t = [q_t^1, \cdots, q_t^N]$ (*cf.* Eq. 8), to achieve vivid and diversified interpretations of $G$. In Table 4c, we present evaluation scores with different numbers of generated target embeddings, *i.e.*, $N = 1, 3, 5, 7$. As can be seen, diversified target parsing indeed boots navigation performance.

**Multitask Planner.** Several variants of multitask planner $f_{\text{MTP}}$ (*cf.* Eq. 9) are compared in Table 4d. The two-layer shared trunk design is adopted, due to its relatively better performance.

**Reward Function.** Next we examine the design of our reward function (§4.4). As seen in Table 4e, each reward term is indeed useful and combining all the four terms leads to the best performance.

**Multitask Learning.** Table 4f reveals the value of training VIENNA on multiple tasks: training with more navigation tasks improves both performance and generalizability. Compared to a composition of four single-task models, multi-task VIENNA also greatly reduces the model size: 101M→31M.

## 6 Conclusion

In this work, we present VXN, a large-scale 3D indoor dataset for multimodal, multitask navigation in continuous and audiovisual complex environments. Further, we devise VIENNA, a powerful agent that simultaneously learns four famous navigation tasks within a single unified parsing-and-query scheme. We empirically show that, through a fully attentive architecture, VIENNA is able to mine and utilize cross-task knowledge to enhance the performance on all the tasks. These efforts move us closer to a community goal of general-purpose robots capable of fulfilling a multitude of tasks.

**Acknowledgement.** This research is partially supported by China National Key R&D Program (2021YFB3101900). Wenguan Wang acknowledges partial support from Australian Research Council (ARC), DECRA DE220101390.

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
