# Towards Versatile Embodied Navigation
## *Supplementary Material*

**Hanqing Wang**[1,2]     **Wei Liang**[1,4*]     **Luc Van Gool**[2]     **Wenguan Wang**[3*]

[1]Beijing Institute of Technology  [2]Computer Vision Lab, ETH Zurich
[3]ReLER, AAII, University of Technology Sydney
[4]Yangtze Delta Region Academy of Beijing Institute of Technology, Jiaxing

Project page: https://github.com/hanqingwangai/VXN

This document provides more details of our approach and additional experimental results, organized as follows:

## I  Experimental Details

**Multitask Distributed Proximal Policy Optimization.** We propose MDPPO based on PPO [1] that is an on-policy RL algorithm with actor-critic architecture. The pseudo code of our training procedure is shown in Algorithm I.

**Training Details.** We apply max clip gradient normalization to all models to stabilize the training. To encourage the agent to explore different actions in reinforcement learning, we add the entropy of predicted action distribution to the loss as a regularizer. We observe that the target position in most episodes can be reached within $50$ steps following the shortest path, hence we set the length of the rollout sequence to $100$ steps during the frame collection, which allows extra exploration steps. For each round of frame collection, we update the agent for $2$ epochs. We use $32$ NVIDIA RTX 2080 GPUs to train the agents. Each GPU runs $4$ training processes. It results in $128$ training processes in total. In single task learning, all $32$ GPUs are assigned to one task. In multi-task learning, each task is assigned with $8$ GPUs. The format of visual observations in the four tasks are identical, where the vertical FOV of the camera is $90°$, the size of RGB observation is $224 \times 224$, and the size of depth observation is $256 \times 256$. In *image-goal nav.* task, the size of the goal image is $224 \times 224$. In *audio-goal nav.* task, the sampling rate of the binaural audio is 16000Hz. Following [2], we transform the binaural audio wave to $41 \times 44 \times 2$ spectrogram map through Short-time Fourier transform. The hyperparameters used in the quantitative experiments are listed in Table. I. We utilize these hyperparameters across the learning of VIENNA$_{\text{ST/MT}}$ and Seq2Seq$_{\text{ST/MT}}$.

**Network Architecture Details.** For fair comparison, the following encoders are used to encode sensory observations and target instructions for Seq2Seq and VIENNA in experiments:

- **Visual Encoder.** Following [3–5], for RGB observation, we apply an ImageNet [6]-pretrained ResNet50 [7] where the last linear classifier is removed to extract semantic visual features. Similarly, for depth observation, we use a modified ResNet50 pretrained in point-goal navigation. The pretrained ResNet50 backbones are frozen during training. Learnable spatial embeddings are concatenated to the visual features to encode spatial features.

---

[*]Corresponding authors.

36th Conference on Neural Information Processing Systems (NeurIPS 2022).

**Algorithm I** The pseudo code of Multitask Distributed Proximal Policy Optimization (MDPPO).

```python
class MDPPOLearner:
    def __init__(self, config, Q: queue):
        self.p = initiates_policy(config) # initiates the policy
        self.gp = self.p.share() # global policy copy
        self.config = config
        self.local_rank = config.local_rank # rank 0 is the host
        self.task_id = self.local_rank%config.task_num # task_id is in [0, 3]
        self.Q = Q # the queue for gradient synchronization

    def train(self):
        for i in range(self.config.max_iters):
            rollouts = self.p.rollout(self.task_id) # frame collection
            for e in range(self.config.epoch):
                self.p.optimizer.zero_grad()
                loss = self.p.PPO_loss(rollouts) # compute PPO loss
                loss.backward()
                self._accumulate_grad()
                if self.local_rank == 0: # if the current process is the host
                    self._average_grad()
                for p, p_g in zip(self.p.parameters(), self.gp.parameters()):
                    p.data = p_g.data.clone()

    def _accumulate_grad(self):
        grads = []
        for p in self.p.parameters():
            grads.append(p.grad)
        self.Q.push(grads)

    def _average_grad(self):
        self.gp.optimizer.zero_grad()
        num = 0
        while not Q.empty() or num < self.config.tasks_num: # sum grads
            num += 1
            grads = Q.get()
            for p, g in zip(self.gp.parameters(), grads):
                p.grad += g
        for p in self.gp.parameters():
            p.grad = p.grad / num
        self.gp.optimizer.step()
```

- **Language Encoder.** Following [3, 5, 8–10], we apply a bi-directional LSTM with pre-trained word embeddings to encode language instructions. Note that in Seq2Seq, only the first and the last hidden states are used to describe the language instruction, while VIENNA utilizes transformer encoders to process the encoded language sequence.
- **Audio Encoder.** Following [2, 11], we apply a CNN of conv $8 \times 8$, conv $4 \times 4$, conv $3 \times 3$ and a linear layer, interleaved with ReLU to encode the spectrogram of binaural audios.

In Seq2Seq, those embeddings are concatenated together and fed into an LSTM planner to encode the episode history. Especially, for multi-task learning, the absent embeddings are padded with zeros. A linear layer is used as the critic to estimate the value based on the hidden state of the LSTM planner. Our proposed VIENNA is viewed as the actor in the training. The embeddings of multimodal sensory observations are first integrated through a transformer encoder (Eq.4) and then fed into the episode history encoder (Eq.6) to obtain contextualized history representation. The augmented instruction embeddings are queried by the averaged episode contextualized history encoding to get the time-varying goal description (Eq.7, 8). Finally, a transformer-based multitask planner consumes the goal description as the query sequence and the episodic history as the key&value sequence to obtain the context for decision making (Eq.9). We average the retrieved context $C_t$ of the multitask planner (see

Table I: The hyperparameters used in the experiments.

| Hyperparameter | Value | Hyperparameter | Value |
|---|---|---|---|
| $N_V$ | 12 | Action entropy coefficient | 0.01 |
| $N_A$ | 12 | Max gradient norm | 0.2 |
| $N_I$ | 16 | Batch size (per process) | 1 |
| $N_L$ | 120 | Number of epochs (per rollout) | 2 |
| $N_G$ | 120 | Number of processes (per GPU) | 4 |
| $d$ | 512 | Total number of GPUs | 32 |
| $r_{\text{success}}$ | 2.5 | Success radius | 1m |
| $r_{\text{slack}}$ | -0.001 | Audio sampling rate | 16000Hz |
| $\gamma$ | 0.99 | Binaural audio spectrogram map size | [41, 44, 2] |
| $\tau$ (PPO) | 0.95 | Camera vertical FOV | $90°$ |
| Learning rate | 0.00025 | RGB observation size | [224, 224] |
| Rollout length | 100 | Depth observation size | [256, 256] |
| Value loss coefficient | 0.5 | Goal image size | [224, 224] |

§4.3) and map it to a scalar as the estimate $\hat{V}_t$ through a linear layer, *i.e.*, $\hat{V}_t = F_{\text{AVG}}(\boldsymbol{C}_t)\boldsymbol{W}^s$. The linear is shared across the tasks.

## II   Audio Simulator

**Continuous Binaural Audio Simulation.**   In [12], chen *et al.*sample a grid of $N$ locations in an environment and simulate the acoustics of the environment by pre-computing binaural room impulse response (BRIRs) of four azimuth angles (*i.e.*, $0°$, $90°$, $180°$ and $270°$) for each possible source and listener placement at the sampled locations. Therefore, the agent is only allowed to move to its adjacent locations. To simulate continuous auditory scenes, we use [13] for real-time BRIRs interpolation. Specifically, we first pre-compute the Dynamic Time Wrapping [14] for each adjacent azimuth angle and location to temporally align BRIRs of different orientations and positions.

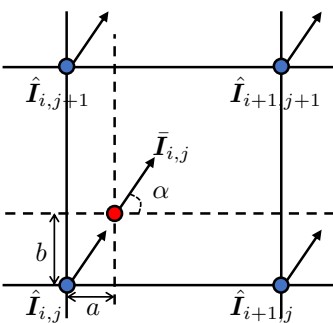

The left channel and right channel of BRIRs are processed respectively. Then we adopt linear interpolation for orientation and bilinear interpolation for position to get BRIRs at an arbitrary location with arbitrary orientation. As shown in Fig. I, the linear interpolation for orientation is computed as:

$$\hat{\boldsymbol{I}}_{i,j} = \frac{2\alpha}{\pi}\boldsymbol{I}_{i,j,0} + (1 - \frac{2\alpha}{\pi})\boldsymbol{I}_{i,j,1}, \tag{I}$$

where $\alpha \in [0, \frac{\pi}{2}]$, $\boldsymbol{I}_{i,j,0}$ and $\boldsymbol{I}_{i,j,1}$ are the temporally aligned BRIRs of adjacent azimuth angles. The final BRIRs $\bar{\boldsymbol{I}}_{i,j}$ is computed as:

Figure I: Interpolation for BRIRs at arbitrary location and orientation.

$$\begin{aligned}\bar{\boldsymbol{I}}_{i,j} =& b(a\hat{\boldsymbol{I}}_{i,j} + (1 - a)\hat{\boldsymbol{I}}_{i+1,j}) \\ &+ (1 - b)(a\hat{\boldsymbol{I}}_{i,j+1} + (1 - a)\hat{\boldsymbol{I}}_{i+1,j+1}),\end{aligned} \tag{II}$$

where $a, b \in [0, 1]$. Some examples of the binaural audio interpolation are available in the video[2].

**Episodic Background Sound Rendering.** Our simulator provides background sound rendering for *image-goal nav.*, *audio-goal nav.*, and *vision-language nav.* tasks. As stated in the main context, the background sound reveals the geometry of the environment, complements the visual cues, and makes the situation closer to the real world. To this end, for each episode, we randomly sample an object within 10 metres of the starting point as the background sound source. As the environments in Matterport3D [15] are densely annotated with semantic labels, we can easily access the locations of objects. To keep the audio semantics consistent with *audio-goal nav.* and *object-goal nav.*, we apply

---

[2]https://youtu.be/Nd1XWCh2r0A

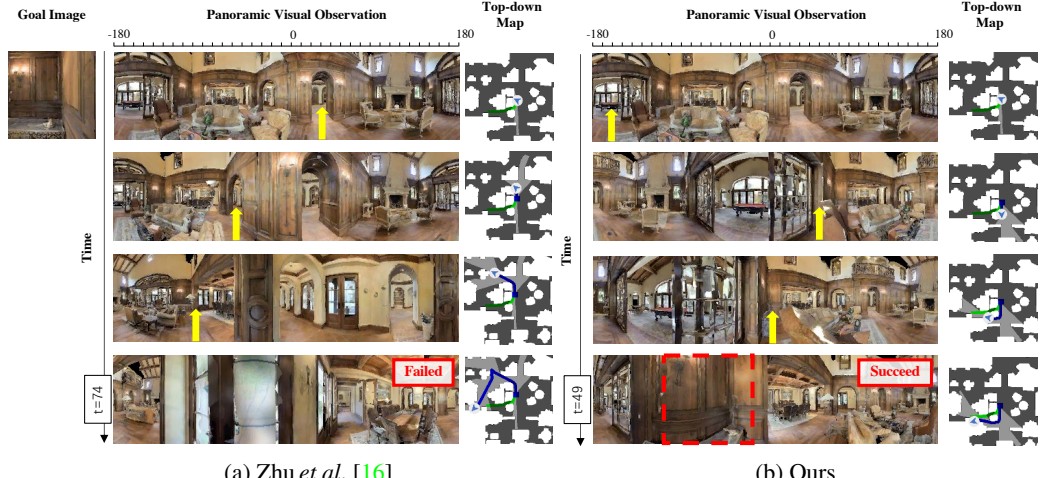

Figure II: A qualitative comparison on *image-goal nav.* task. (a) is the trajectory of Zhu *et al.* [16] and (b) is the trajectory of our method. The *green* trajectory in the top-down map is the shortest path to the goal. The *blue* trajectory in the top-down map is the path that the agent navigates. The dashed box in (b) shows the corresponding area in the goal image.

the sounds of 21 object categories following [11] as the background sounds. It is worth noting that the sound source is randomly picked for each time an episode starts during training. In the evaluation phase, the background sound source is randomly sampled and then fixed for the same episode. Some examples are available in the video[3].

## III Qualitative Results

**Qualitative Comparison.** We present some additional qualitative results of our approach and the specifically designed methods for the four tasks. Fig. II shows the comparison with Zhu *et al.* [16] on *image-goal nav.*. In this case, the agent is asked to reach a corner with a table and a lamp on the wall. Our agent is able to conduct more robust navigation compared to [16]. Fig. III shows the comparison with Chen *et al.* [12]'s model on *audio-goal nav.*. In this case, the sound source is the noise of swinging a towel in the bathroom. Our agent is able to locate the sound source accurately and navigate toward it. The comparison with Chaplot *et al.* [17] on *object-goal nav.* is illustrated in Fig. IV, the goal object category is '*table*'. Our agent can consistently locate the semantic instance in panoramic visual observation. Fig. V shows the comparison with Krantz *et al.* [3] on *vision-language nav.* task. In this case, our agent is able to consistently navigate following the instruction.

## IV Discussion

We present VXN, a realistic multitask navigation dataset on a publicly available interior dataset Matterport3D [15]. It naturally combines four classic navigation tasks in standardized continuous audiovisual-rich environments, which provides a foundation for developing a versatile embodied navigation agent. This new setting is much more challenging compared to any of those single tasks in two main aspects: 1) With more modalities involved, the agent needs to do multimodal co-grounding and inference accordingly, which remains an open problem in the field. 2) The instructions of different navigation tasks are detailed in different granularity, which requires different navigation policies. For example, the object category tag in *object-goal nav.* suggests the agent for going somewhere that contains the goal object instances and finding one. Based on the tag, the agent needs to explore the environment. Nevertheless, the language instruction in *vision-langauge nav.* is an explicit instruction that guides the agent to navigate step by step. In this case, instruction fidelity is important to the navigation policy. As revealed in our quantitative experiments, the current methods still have a large room for improvement on all those tasks while our multitask agent first explores the inherent relations between the fine-grained navigation tasks and makes progress. We believe that mastering a group

---

[3] https://youtu.be/_TR24NAc92M

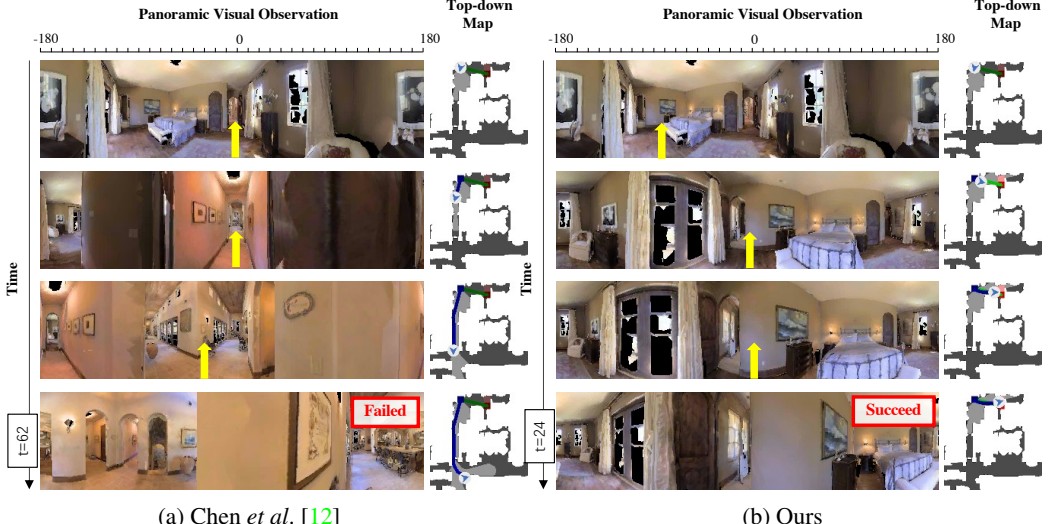

(a) Chen *et al*. [12]  (b) Ours

Figure III: A qualitative comparison on *audio-goal nav.* task. In this case, the sound source is the noise of swinging a towel in the bathroom. (a) is the trajectory of Chen *et al*. [12] and (b) is the trajectory of our method. The *green* trajectory in the top-down map is the shortest path to the goal. The *blue* trajectory in the top-down map is the path that the agent navigates.

**Goal Object:** *table*

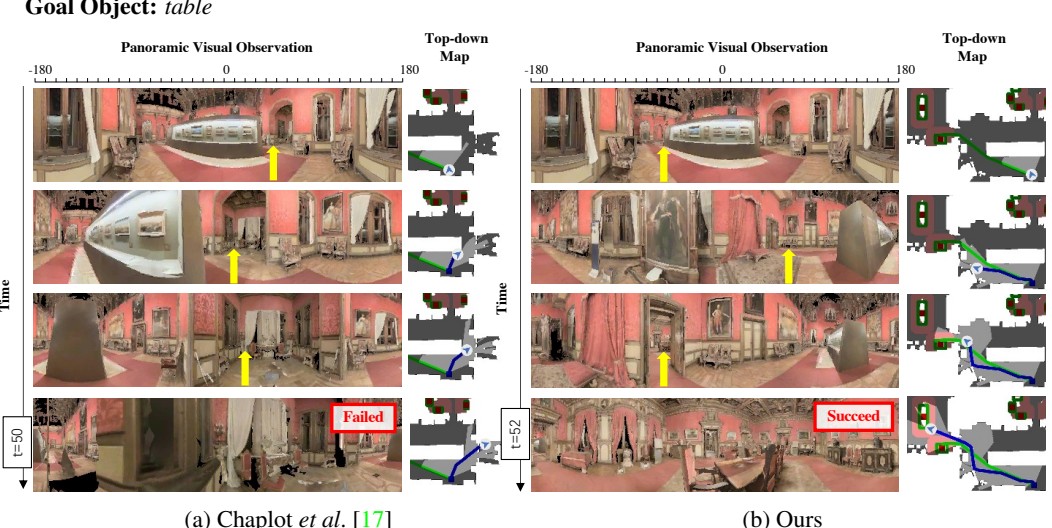

(a) Chaplot *et al*. [17]  (b) Ours

Figure IV: A qualitative comparison on *object-goal nav.* task. The goal object category is '*table*'. (a) is the trajectory of Chaplot *et al*. [17] and (b) is the trajectory of our method. The *green* trajectory in the top-down map is the shortest path to the goal. The *blue* trajectory in the top-down map is the path that the agent navigates.

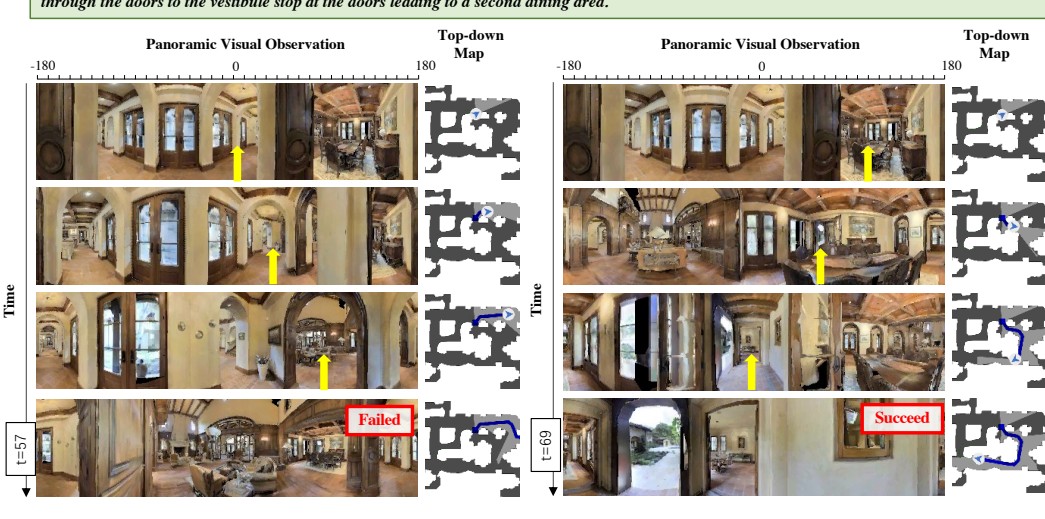

> *Instruction: Go through the wooden doors to the dining area. Walk past the dining table to the glass and wood doors beyond. Turn right and go through the doors to the vestibule stop at the doors leading to a second dining area.*

(a) Krantz *et al*. [3]         (b) Ours

Figure V: A qualitative comparison on *vision-language nav.* task. (a) is the trajectory of Krantz *et al*. [3] and (b) is the trajectory of our method. The *blue* trajectory in the top-down map is the path that the agent navigates.

of fine-grained navigation tasks through a powerful universal agent is more elegant and potentially promising. It will facilitate future work in this field.

**Limitations & Societal Impact.** Our real-time binaural audio rendering relies on the pre-computed binaural room impulse responses provided by [12]. For some navigable locations where the adjacent BRIRs are unavailable (due to the coarse size of grid sampling in [12]), we use the nearest BRIRs as an approximation. We observe some flaws of the audio rendering that the interpolated binaural audio slightly jitters sometimes. It is probably caused by the dynamic time wrapping of different BRIRs pairs. A possible workaround to alleviate the jitter is to smooth the interpolated audio through a low-pass filter. Compared to the Seq2Seq models, our transformer-based model is more memory-consuming during inference. This can be partially solved by introducing a sliding memory buffer for history observations similar to [11]. The navigation agents are developed in virtual simulated environments. If the algorithm is deployed on a real robot in a real dynamic environment, the collisions during navigation can potentially cause damage to persons and assets. More work should be done to practice real-world deployment, *e.g.*, introducing hard constraints to the action space to avoid collisions, and including additional experiments to study the risk of potential damage.