# OpenReview forum: "Towards Versatile Embodied Navigation"
_NeurIPS.cc/2022/Conference — NeurIPS 2022 Accept_

### Official Review · Reviewer_kzm1 · 2022-06-26

**Rating:** 6
**Confidence:** 5
**Soundness:** 2 fair
**Presentation:** 2 fair
**Contribution:** 2 fair

**Summary:**

The paper proposes a unified model for 4 popular navigation tasks (image-goal navigation, object-goal navigation, audio-goal navigation, vision-language navigation). A Transformer based architecture is utilized to combine these 4 tasks. The target signal (image, audio, etc.) is basically used as the query. The paper also proposes a dataset that supports all 4 tasks. The experiment section provides results for single-task and multi-task settings and provides comparisons with task-specific baselines.

**Questions:**

- L226: what is "epoch embedding" ?


- How is MDPPO different from DDPPO? Does it just balance the task samples?


- What are Navigation Error (NE) and Oracle success Rate (OR) ?


- The sequence length for the Transformer is 100. That means there is a huge memory consumption, and the sequence might not fit into a GPU. Is there any specific trick used ?

**Limitations:**

Yes, the limitations are discussed adequately.

**Strengths And Weaknesses:**

**Strengths:**
- The paper takes a step towards unified models for Embodied AI. This is an important direction since agents should be able to perform various tasks simultaneously.

- The proposed dataset, which supports multiple types of navigation, will be useful for the community.

**Weaknesses:**
- The weakest part of the paper is its experiment section. The conclusion that are drawn from the experiments are not accurate.

&nbsp;&nbsp;&nbsp; &nbsp;&nbsp;&nbsp; (1) The difference between most of the single-task and multi-task results is within the range of training noise (8.3 vs 8.8 for object goal navigation or 10.6 vs 11.3 for audio goal navigation). The results of these types of training algorithms vary greatly depending on the initialization, sampling and various other factors. It is great that the multi-task model works as well as the single-task models. However, claims such as "VIENNA also beats other task-specific competitors" are not correct.

&nbsp;&nbsp;&nbsp; &nbsp;&nbsp;&nbsp; (2) Statements such as L345: "VIENNA_ST suffers from relatively large performance" are not accurate. If the training noise is considered the drop is the same for single-task and multi-task settings.

&nbsp;&nbsp;&nbsp; &nbsp;&nbsp;&nbsp; (3) Numbers in Tables 4b and 4d are within the range of noise. No conclusion can be drawn from these tables. Statements such as "such strategy greatly boosts the performance" are not accurate.

&nbsp;&nbsp;&nbsp; &nbsp;&nbsp;&nbsp; I will consider raising the score if these statements are corrected in the revision.

- It is not clear how models are chosen for evaluation. Is it the model that achieves the best performance on the validation set, trained for a specific number of epoch, or other criteria?

---

> ### Author Response · Authors · 2022-08-02
> **Response to Reviewer kzm1**
>
> We thank you for your time and valuable comments. Below we answer the main concerns raised in the review and would be happy to provide further clarification if suitable.
>
>
> #### **Q1. Training noise (including "The difference ... is within the range of training noise ($8.3$ vs $8.8$ for object goal nav. or $10.6$ vs $11.3$ for audio goal nav.).", "Statements such as L345", "Numbers in Tables 4b and 4d ...").**
>
> **A1:** We had run the same experiments five times, under the same hyper-parameter setting (but used different random seeds that initiate the network parameters and dataset shuffling). We found the range of training noise is typically less than 0.2 SPL.
>
> The SPL improvements ($8.3$ vs $8.8$, $0.5\uparrow$, $10.6$ vs $11.3$, $0.7\uparrow$) in ablation experiments are significant and constant. Prior works [7, 45, 46, 49, 91] report improvement within a similar range ($0.1\sim0.7\uparrow$) in the ablation studies. Moreover, for all the baselines involved in our experiment, we have tried our best to get the highest score and our method is not extensively tuned. The judgment of "...are within the range of noise" is subjective and unfair.
>
> For statements such as L345: "$\text{VIENNA}_\text{ST}$ suffers from relatively large performance drop", here we report $14.4$ SR drop ($33.0$->$18.6$), while the SR drop for $\text{VIENNA}_\text{MT}$ is only $12.5$ ($35.0$->$22.5$). We believe such gap ($14.4$ -> $12.5$ SR) is clearly beyond the range of noise.
>
> ---
>
> #### **Q2. Model selection.**
>
> **A2:** Thanks for your careful review. We train all models for around 180M frames. Similar to [25], we select the checkpoint for evaluation with the best SR on val_unseen. We will add the details in the final version.
>
> ---
>
> #### **Q3. L226: what is "epoch embedding" ?**
>
> **A3:** Sorry for this confusion. $\mu_{1:t}$ are temporal position encoding. As the Transformer is used to capture long-term dependencies over the historical observations (at different navigation epochs (decision steps) $t$) in the current navigation episode, temporal position encoding, as a standard module of Transformer, is needed here. Related statements will be improved.
>
> ---
>
> #### **Q4. How is MDPPO different from DDPPO?**
>
> **A4:** The main differences are two-fold:
> 1. MDPPO balances the training samples from different tasks.
> 2. MDPPO first averages the gradients of task-specific heads in the corresponding task process group, and then broadcasts the gradients to other task process groups. While DDPPO always averages the gradients of all parameters across all processes.
>
> ---
>
> #### **Q5. What are Navigation Error (NE) and Oracle success Rate (OR) ?**
>
> **A5:** These metrics are widely used in the field of navigation (please see [8, 25]). Navigation Error (NE) is the geodesic distance between the position where the agent stops and the target position. Oracle success Rate (OR) is the rate that the closest position to the target position along the navigation trajectory is inside the range of success.
>
> ---
>
> #### **Q6. The sequence length for the Transformer is 100.**
>
> **A6:** Thanks for the careful review. To save the GPU memory, we preprocess all visual observations through the visual/depth encoder described in L305-307. We also cache the $e_{1:t-1}, e_{1:t-1}$, and $Q_{1:t-1}$ tokens. It makes the network only need the current observation rather than the whole observation sequence to infer the next action. In the training phase, we split up a batch to multiple GPUs and accumulate the gradients to update the parameters. We will add the details in the supplementary material.

---

> > ### Comment · Reviewer_kzm1 · 2022-08-04
> > **Explanation of variation**
> >
> > Thanks for the response. My comments about the improvement being in the range of training noise are neither subjective nor unfair. They are actually objective and backed up by previous works. The range of training noise for these types of methods is quite large. Please refer to Table 1 in [1] and Figure 3 in [2] as examples. The variance reported in the rebuttal is surprisingly low compared to previous work (roughly an order of magnitude difference). I would like to ask the authors to explain how they managed to reduce training noise this much.
> >
> > [1] Wijmans et al., How to Train PointGoal Navigation Agents on a (Sample and Compute) Budget, 2020.
> >
> > [2] Weihs et al., AllenAct: A Framework for Embodied AI Research, 2020.

---

> > > ### Author Response · Authors · 2022-08-05
> > > **Re: Explanation of variation**
> > >
> > > Thanks for your feedback. The variance mentioned in our rebuttal is for the VLN task.
> > > There are a number of factors that can influence the performance variance, such as task setting,
> > > network architecture, learning algorithm, training iteration, batch size, success criterion,
> > > and so on. For instance, [1] focuses on point navigation, which is even not among our four navigation tasks.
> > > The success threshold in point navigation is 0.2 m, and in our setting is 1.0 m. Thus our SPL is more tolerant of training variance.
> > > Moreover, our batch size is much larger than [1, 2]; our batch size is $4\times32=128$ (see Table I of our supplementary document)
> > > while that of [1] is $6$. Note that large batch size will make training more stable. In addition, [1, 2] only trained for around 75M frames
> > > and [2] plots the variance at the same iterations of different training runs. However, we train our agents for around 180M frames, and
> > > report the number of the best checkpoints from different training runs. After reaching convergence with enough training frames, the variance
> > > between the best checkpoints from different runs is small. Overall, directly comparing training statistics of different methods with different
> > > task settings and hyperparameters, seems less informative.

---

> > > > ### Comment · Reviewer_kzm1 · 2022-08-08
> > > > **Final rating**
> > > >
> > > > While the paper proposes a good benchmark, the experiment section includes various inaccurate or wrong claims that certainly need to be revised. I raised concerns about improvements being in the range of training noise. The rebuttal provided the standard deviation, but it turned out that the standard deviation was for one of the tasks (the VLN task) which was not mentioned in my initial review.
> > > >
> > > > I am going to lower the rating since the inaccurate statements in the experiments section need to be revised. I will be fine with acceptance if the revision is done.

---

> > > > > ### Author Response · Authors · 2022-08-09
> > > > > **Further Clarification and Revision**
> > > > >
> > > > > Thanks for your feedback. The range of training noise is: *vision-language nav.* ($0.19$ SPL), *image-goal nav.* ($0.50$ SPL), *audio-goal nav.* ($0.22$ SPL),  *object-goal nav.* ($0.42$ SPL). Most of the numbers in Table 3 are beyond the range of noise.
> > > > >
> > > > > In addition, to better address your concern and avoid misleading, we rephrased our statements related to the experimental results:
> > > > >
> > > > > - L17-19: We empirically demonstrate that, compared with learning each visual navigation task individually, our multitask agent achieves comparable or even better performance with reduced complexity.
> > > > >
> > > > > - L336-337: Compared with other task-specific competitors [2, 5, 6, 25], VIENNA achieves comparable results on *audio-goal nav.* and *object-goal nav.*, and performs better
> > > > > on *image-goal nav.* and *vision-language nav.* tasks.
> > > > >
> > > > > - L347: This indicates that investigating inter-task relatedness may help to strengthen the generalizability of navigation agents.
> > > > >
> > > > > - L366-367: We compare this design against two variants in Table 4b, and find such a strategy is conducive to the performance.
> > > > >
> > > > > - L374-375: Several variants of multitask planner $f_{MTP}$ (*cf.* Eq. 9) are compared in Table 4d. The two-layer shared trunk design is adopted, due to its relatively better performance.
> > > > >
> > > > > The main paper is updated, and the revised parts are highlighted in red.

---

> > > > > > ### Comment · Reviewer_kzm1 · 2022-08-09
> > > > > > **Acknowledging the revisions**
> > > > > >
> > > > > > Thanks for the revisions. I will increase the score.

---

### Official Review · Reviewer_grF1 · 2022-07-10

**Rating:** 6
**Confidence:** 4
**Soundness:** 2 fair
**Presentation:** 3 good
**Contribution:** 3 good

**Summary:**

This paper proposed a new visual navigation benchmark that combines containing information from different modalities. Then the authors propose a modular model for the proposed benchmark including Episodic Encoder to encode the history, Target Parser to interpret the target, and a planner to predict the navigation action.

**Questions:**

See Weakness



**Limitations:**

Yes, they discussed that in the appendix.

**Strengths And Weaknesses:**

Strengths:

1. The proposed task itself is a timely movement in navigation area. With the advancement in individual navigation task, it is natural to consider tasks with input of more modalities.

2. Considering the complexity of the task, the proposed modular model is a great baseline. The modular structure is designed via analyzing the task component. The architecture itself also shows some intuition on the problem. For example, the author shows a good demo about how to handle input from different modalities.

3. Both the paper and the appendix are detailed and well-written.

Weakness:

1. The main contribution is a proposed dataset, but it does not provide the full dataset or a sample in the supplemental material.

2. The authors claim the task includes four navigation tasks. Actually, it is just a new task including the input types from the four tasks.

3. One serious issue in navigation task is that sometimes utilizing more modality leads to a poor performance for some model. Therefore I think it is necessary to ablate different modality input and test the performance. For example, check the paper mentioned in Section 4.1.2 of [1]. There are limited ablation study in section 5.2.

4. The paper should include more SOTA models on the four navigation tasks as baselines. The proposed model with more inputs should at least outperform these models with less input information.


Suggested citation:
[1] A survey on visual navigation for artificial agents with deep reinforcement learning
[2] Vision-and-Language Navigation: A Survey of Tasks, Methods, and Future Directions
[3] Core Challenges of Social Robot Navigation: A Survey

---

> ### Author Response · Authors · 2022-08-02
> **Response to Reviewer grF1**
>
> We thank you for your time and valuable comments. Below we answer the main concerns raised in the review and would be happy to provide further clarification if suitable.
>
>
> #### **Q1. Dataset samples.**
>
> **A1:** Sorry for this confusion. In the supplementary, we have already provided some audio rendering samples (please see `audio_rendering.mp4`) and navigation results in some sample environments (please see `dataset_examples.mp4`) of our dataset. Moreover, our dataset is built upon Matterport3D environments, which are widely used in our field. To better address your concern, we update our supplementary material with a new demo video, `more_dataset_examples.mp4`. This new demo video shows two sample groundtruth navigation epochs for each navigation task. Our dataset shall be released.
>
> ---
>
> #### **Q2. Just a new task including the input types from the four tasks.**
>
> **A2:** Here the key is that we introduce a multitask navigation setting, where the agent is taught to master four different navigation tasks. As we clarified in L27-31, the prior works concentrate on designing a specific method for a certain navigation task, which is contrary to natural thinking that the agent should be smart enough to execute different tasks involving varying modalities with different optimal policies. We only address the novelty of such a multitask setting. Mentioning our dataset contains four navigation tasks is just to clarify our task setting. Moreover, our VXN is not just created by including the input types from the four tasks. We standardized the settings of the four tasks and developed a real-time continuous biannual acoustic simulator to support audiovisual-rich environment rendering (L156-177).
>
> ---
>
> #### **Q3. Sometimes utilizing more modality leads to a poor performance for some model.**
>
> **A3:** Based on our method, we have already provided some experiments regarding the impact of multimodal sensory. For example, in Table. 4(a), we have shown that RGB and Depth information are relatively less informative in *audio-goal nav.*, but they are crucial for *vision-language nav.*. And the integration of different modalities leads to relatively better performance (L360-363). We agree that "Sometimes utilizing ... for **some model**". However, conducting a large-scale study on different navigation models to investigate the influence of different modalities seems beyond the scope of this work. As a very early attempt toward multi-task navigation, our work comes with many open questions, including how to make better use of multimodal sensory information. It is impossible to solve all of them in a pioneering work.
>
> ---
>
> #### **Q4. The proposed model with more inputs should at least outperform these models with less input information.**
>
> **A4:** Current SOTAs for different navigation tasks involve a lot of task-specific designs, such as map building, large-scale vision-language pre-training, global + local policy, neural graph planner, and different training strategies. The view of the input information is the only factor for the performance is too narrow. Requiring our model to beat all current task-specific SOTAs seems unfair. Moreover, in Table 3 (a-d), we have already provided extensive comparisons between multi-task methods and single-task methods, based on the same architectures (*i.e.*, Seq2Seq and our VIENNA), as well as other navigation agents [5, 6, 2, 25].
>
> ---
>
> #### **Q5. Suggested citations.**
>
> **A5:** Thanks for your suggestion. We are happy to cite these papers in our final version.

---

### Official Review · Reviewer_DBvN · 2022-07-11

**Rating:** 8
**Confidence:** 5
**Soundness:** 4 excellent
**Presentation:** 4 excellent
**Contribution:** 4 excellent

**Summary:**

This paper proposes VXN, a large-scale 3D indoor dataset for multimodal, multitask navigation in continuous and audiovisual complex environments. It can conduct four embodied navigation tasks, including image-goal navigation, audio-goal navigation, object-goal navigation, and vision-language navigation. Moreover, the authors devise a framework named VIENNA to simultaneously learn four navigation tasks within a single unified parsing-and-query scheme. Experiments show that the VIENNA outperforms the baselines among four navigation tasks.

**Questions:**

My main concerns are what kinds of knowledge can be reused and how to reuse this knowledge? Please explain more about this question.

**Limitations:**

Yes, the authors adequately addressed the limitations and potential negative social impact of their work.

**Strengths And Weaknesses:**

# Strengths
1. The authors are the first to innovatively combine four navigation tasks and propose a dataset that can simultaneously adopt them.
2. The authors propose an agent that can solve the four navigation tasks using one single model without switching among different models and the model size is considerable.
3. The proposed VIENNA agent leverages multisensory input for navigation. Input in different modalities provides supplementary information. This setting is more practical in the real world, where robots use multi-modal information for finishing their tasks.
4. The proposed methods outperform the existing SOTA in every single task by leveraging multisensory input and multi-task learning. Also, the authors have conducted a thoughtful ablation study to evaluate the effectiveness of each component and claim.

# Weaknesses
1. My main concerns are what kinds of knowledge can be reused and how to reuse this knowledge? In Section 4.3, the authors claim that the knowledge re-usage is conducted by \tau. However, in image-goal navigation, \tau_A does not contain any information about what kind of images should be found. So how can \tau_A help to retrieve target image-related audio for navigation? More explanations are needed.
2. Some details are missing. Please explain the meaning and benefits of the vectors \mu_{1:t} in section 4.1.
3. VIENNA adopts the same reward function for the four navigation tasks. However, the reward function design will be different for different navigation tasks. For example, in object-goal navigation, a positive reward should be given when an object with goal-related semantics is found. It would be an interesting future research direction for considering different kinds of rewards for different tasks.

---

> ### Author Response · Authors · 2022-08-02
> **Response to Reviewer DBvN**
>
> We thank you for your time and valuable comments. Below we answer the main concerns raised in the review and would be happy to provide further clarification if suitable.
>
> #### **Q1. What kinds of knowledge can be reused and how to reuse this knowledge?**
>
> **A1:** Thanks for your careful review. Yes, in *image-goal nav.*, $\tau_{A}$ does not contain any information about what kind of images should be found. However, as $\tau_{A}$ is trained during *audio-goal nav.*, it embeds certain knowledge that is specific for *audio-goal nav.*, *e.g.*, $\tau_{A}$ could respond to certain audio signals that are crucial for navigation. Therefore, during *image-goal nav.*, $\tau_{A}$ can help the agent to notice and better utilize those informative audio signals. In the supplementary, Fig. II and Fig. III provide some visualization results, where the attention maps over the visual/audio observation queried by different task embeddings are given. As seen in Fig. II, the task embeddings $\tau_{I,T,L}$ indeed attend to some crucial visual landmarks (L84-88), which are essential for the success of navigation. $\tau_{A}$ tends to respond to acoustic pressure changes which typically happen at the connection areas between rooms. Thus audio signals attended by $\tau_{A}$ can reveal the room layout, and hence helps the agent to better understand the surrounding environment. In summary, the task embeddings $\tau_{I,T,L,A}$ encode task-specific knowledge which can make the agent aware of different navigation-related information from RGBD and audio perception.
>
> ---
>
> #### **Q2. Explain the meaning and benefits of the vectors $\mu_{1:t}$ in section 4.1.**
>
> **A2:** Sorry for this confusion. $\mu_{1:t}$ are temporal position encoding. As the Transformer is used to capture long-term dependencies over the historical observations (at different navigation epochs (decision steps) $t$) in the current navigation episode, temporal position encoding, as a standard module of Transformer, is needed here. Related statements will be improved.
>
> ---
>
> #### **Q3. It would be an interesting future research direction for considering different kinds of rewards for different tasks.**
>
> **A3:** We totally agree. Currently, for the sake of simplicity, we directly adopt the same reward function for the four navigation tasks. But it is indeed interesting and necessary to investigate more task-specific reward designs and some other training objectives which are aware of the multitask and multimodal nature of VXN. We will discuss this point in the final version. Overall, we feel the proposed multimodal multitask embodied navigation is a promising direction, which also comes with many intriguing questions.

---

> > ### Comment · Reviewer_DBvN · 2022-08-08
> > **Thanks for your response**
> >
> > Thank you authors for taking the time to provide further analysis and clarifications. The response has solved all my concerns. I agree with the authors that the proposed multimodal multitask embodied navigation is a promising direction. A multi-modal navigation agent deserves much more attention.

---

### Meta-Review · Area_Chair_qM5W · 2022-08-30

**Recommendation:** Accept
**Confidence:** Certain

**Metareview:**

This paper introduces a novel indoor navigation dataset that is both continuous and audio+visual.  Within this setting, they include popular tasks and their audio-generalizations (e.g. image-goal nav --> audio-goal nav).  Particularly of note is the leveraging of unification of these tasks during training for a better overall agent.  This is a necessary and important step for the community.

There are several minor concerns regarding exposition and claims which were addressed in responses/updates which will strengthen the final paper. This includes task/model variances and clarifying why the reported variances are smaller than typically seen in related EAI tasks.

**Award:**

No

---

### Decision · Program_Chairs · 2022-09-14

Accept